# The Integrated Bioinformatic Approach Reveals the Prognostic Significance of LRP1 Expression in Ovarian Cancer

**DOI:** 10.3390/ijms25147996

**Published:** 2024-07-22

**Authors:** Tesfaye Wolde, Vipul Bhardwaj, Md. Reyad-ul-Ferdous, Peiwu Qin, Vijay Pandey

**Affiliations:** 1Institute of Biopharmaceutical and Health Engineering, Tsinghua Shenzhen International Graduate School, Tsinghua University, Shenzhen 518055, China; tesfalem2002@gmail.com (T.W.); rockyreyad@sz.tsinghua.edu.cn (M.R.-u.-F.); 2Tsinghua Berkeley Shenzhen Institute, Tsinghua Shenzhen International Graduate School, Tsinghua University, Shenzhen 518055, China; bhardwajvipul27@yahoo.in

**Keywords:** ovarian cancer, tumour microenvironment, immune-related genes, bioinformatics, LRP1, prognosis

## Abstract

A hyperactive tumour microenvironment (TME) drives unrestricted cancer cell survival, drug resistance, and metastasis in ovarian carcinoma (OC). However, therapeutic targets within the TME for OC remain elusive, and efficient methods to quantify TME activity are still limited. Herein, we employed an integrated bioinformatics approach to determine which immune-related genes (IRGs) modulate the TME and further assess their potential theragnostic (therapeutic + diagnostic) significance in OC progression. Using a robust approach, we developed a predictive risk model to retrospectively examine the clinicopathological parameters of OC patients from The Cancer Genome Atlas (TCGA) database. The validity of the prognostic model was confirmed with data from the International Cancer Genome Consortium (ICGC) cohort. Our approach identified nine IRGs, *AKT2*, *FGF7*, *FOS*, *IL27RA*, *LRP1*, *OBP2A*, *PAEP*, *PDGFRA*, and *PI3*, that form a prognostic model in OC progression, distinguishing patients with significantly better clinical outcomes in the low-risk group. We validated this model as an independent prognostic indicator and demonstrated enhanced prognostic significance when used alongside clinical nomograms for accurate prediction. Elevated *LRP1* expression, which indicates poor prognosis in bladder cancer (BLCA), OC, low-grade gliomas (LGG), and glioblastoma (GBM), was also associated with immune infiltration in several other cancers. Significant correlations with immune checkpoint genes (ICGs) highlight the potential importance of LRP1 as a biomarker and therapeutic target. Furthermore, gene set enrichment analysis highlighted LRP1’s involvement in metabolism-related pathways, supporting its prognostic and therapeutic relevance also in BLCA, OC, low-grade gliomas (LGG), GBM, kidney cancer, OC, BLCA, kidney renal clear cell carcinoma (KIRC), stomach adenocarcinoma (STAD), and stomach and oesophageal carcinoma (STES). Our study has generated a novel signature of nine IRGs within the TME across cancers, that could serve as potential prognostic predictors and provide a valuable resource to improve the prognosis of OC.

## 1. Introduction

Ovarian cancer (OC) represents the most prevalent malignancy among gynaecological diseases globally, accounting for an estimated 3.7% of the incidence and 4.7% of the mortality [1,2]. Despite notable advances in the management of OC patients, outcomes remain complicated [3]. This is primarily due to the lack of distinct symptoms and reliable screening methods, resulting in late diagnosis [4]. Furthermore, OC comprises several subtypes with distinct biological and molecular properties, even within the same histological subtype, and there is inconsistency in the availability and access to treatment [5]. Cytoreductive surgery and chemotherapy using platinum- and taxane-based drugs are the primary treatment options for OC patients [6]. Initially, the success rate of the primary treatment was about 80–90%, with a reasonable response rate in patients with OC [7,8]. However, more than 85% of women who have undergone primary treatment experience recurrence, resulting in recurrent OC within two years of achieving complete remission [9]. Recognizing the challenges facing OC treatment, there is an urgent need for research into potential molecular mechanisms and prognostic markers. This investigation aims to identify new anti-tumour therapy targets that may be significantly associated with the therapeutic response of OC patients [10].

Recent research has demonstrated that tumour heterogeneity and intricate signalling pathways between tumour cells and the surrounding TME are strongly associated with poor outcomes in patients with OC [11,12]. The TME, which is made up of vessels, immune cells, and extracellular matrix (ECM), has been reported to promote cancer growth, invasion, and metastasis [12,13]. Furthermore, recent data from numerous studies have demonstrated that TME cells and the extent of immune and stromal cell infiltration into tumours are significant factors that influence prognosis and anti-tumour response [14,15,16,17]. Immune and stromal cells are the two main types of non-tumour components in the TME and have been proposed as valuable for the diagnosis and prognosis evaluations of tumours [18]. For instance, Huo and associates recently identified ten prognostic-related TME genes and immunity in OC patients [14]. Another study done by Olalekan and colleagues revealed the immune cell types and their roles in the TME of metastatic OC through single-cell transcriptomics [19]. Furthermore, a study performed by Wu et al. demonstrated that Collagen type XI alpha 1 promotes OC growth and invasion by activating cancer-associated fibroblasts (CAFs) [20]. Yu et al. reported that a high expression of CD47 is closely linked to the immune infiltration of OC cells, affecting the TME and potentially inducing tumour heterogeneity [10]. The contact between tumour and stromal cells is regulated bidirectionally by a variety of pharmacological agents, including, taxanes, platinum compounds, and PARP inhibitors, which significantly impact the TME [21,22]. While the interactions between different immune cells and cancer cells within the TME are well understood, further study is needed to fully elucidate the molecular mechanisms underlying the TME regulation of OC progression. A comprehensive understanding of the TME and its heterogeneity will facilitate the development of more effective therapeutic targets for OC patients. Our goal is to identify a prognostic biomarker that could contribute to explaining the TME heterogeneity of OC.

Low-density lipoprotein receptor-related protein 1 (LRP1) is a multifunctional endocytic receptor [23,24] that plays a crucial role in various physiological processes, including lipid metabolism [25], cell signalling, and the elimination of apoptotic cells [26]. Beyond its well-established physiological roles, emerging evidence has highlighted the role of LRP1 in the complex landscape of cancer biology [27,28]. *LRP1* has been found to influence cancer progression and metastasis by regulating cell migration, adhesion, proliferation, and the modulation of the TME [29]. *LRP1* interacts with a myriad of ligands and co-receptors, mediating crosstalk between cancer cells and their surrounding stroma, thus impacting pathways critical for tumour growth and dissemination [30,31]. LRP1 expression and its function in cancer are context-dependent, varying across different types of tumours and stages of disease, making LRP1 both a prospective target for therapeutic intervention and a possible biomarker for cancer prognosis. However, the effect of LRP1 on the TME and prognosis of OC have not been fully investigated. Therefore, a comprehensive approach is necessary to understand the multifaceted roles of LRP1 in cancer, including its contributions to tumour cell behaviour, interaction with the microenvironment, and influence on metastatic potential, which collectively highlights the clinical significance of LRP1 expression in OC progression.

In this investigation, we carried out an exhaustive examination of OC data to investigate the function and heterogeneity of immune-related genes (IRGs) using GEPIA (Gene Expression Profiling Interactive Analysis) and the ImmPort (The Immunology Database and Analysis Portal) database. Several bioinformatics tools, including univariate Cox regression, multivariate Cox regression, and LASSO Cox regression, were utilized to screen out nine differentially expressed IRGs (*LRP1*, *IL27RA*, *FGF7*, *FOS*, *AKT2*, *PAEP*, *OBP2A*, *PI3*, and *PDGFRA*) that demonstrate prognostic and diagnostic value. Among these nine IRGs, LRP1 exhibited the best theragnostic (therapeutic and diagnostic) value. We next examined the association between LRP1 and various factors, such as prognostic value and mRNA gene expression, using Oncomine, GEPIA, TNM, Sangerbox3, CancerSEA, and pan-cancer analyses. Furthermore, we utilized the TIMER database and Sangerbox3 to explore the relationship between immune infiltration levels and *LRP1* expression in various malignancies, including OC. We also assessed the association between ICGs and *LRP1* expression in 39 types of cancer. To further illustrate the co-expression genes of *LRP1* and its regulatory networks, including functional pathways in OC, gene set enrichment analysis (GSEA) was performed. This study aims to discover a novel IRG signature and the role of LRP1 in immune infiltration for OC prognosis. Our research could provide a fresh understanding of LRP1’s role in the OC TME and highlight the potential for the therapeutic targeting of LRP1.

## 2. Results

### 2.1. LRP1, IL27RA, and FGF7 Emerge as Key Cancer-Related IRGs in OC

To identify the subset of IRGs implicated in OC progression, we meticulously analysed data from the Immunology Database and Analysis Portal (ImmPort; https://www.immport.org) accessed on 1 February 2024, following the outlined methodology [32] detailed in the methods and materials section of our study.

This approach identified 7641 genes with differential expression in OC samples compared to adjacent normal ovarian tissues. Of these, 554 genes were immune-related in OC, with 206 downregulated and 348 upregulated (Figure 1A,B).

Enrichment studies examined the biological implications of these 554 differentially expressed IRGs. KEGG pathway analysis revealed significant enrichment in “chemokine signaling”, “Th17 cell differentiation”, and “antigen presentation pathway” (Figure 1C). Gene Ontology (GO) analysis provided details on growth factor receptor binding and cytokine receptor binding pathways (Supplementary information (Appendix A).

We used protein–protein interaction (PPI) network analysis to explore interactions among these proteins in biological systems. With a confidence threshold of 0.95, we identified significant associations, notably involving proteins like LRP1, IL27RA, and FGF7, which showed the highest degrees of association (Appendix A). These findings provide valuable insights for future target gene screening and research into the mechanisms of IRGs in OC progression.

### 2.2. The Nine IRGs Reflect Poor Survival and Tumour Progression in OC

Our goal was to elucidate the potential functional association of nine IRGs with OC survival and tumour progression. We analysed clinicopathological parameters, including tumour pathological stage (*p* = 0.1301) and grade (*p* = 0.6302), and the expression levels of the nine IRGs, shown in a heatmap (Appendix A). Although statistical significance was not observed, likely due to sample size limitations, a trend indicated that high-risk patients tended to have advanced tumour stages and lower grades. Additionally, higher mortality was observed in the high-risk OC group, suggesting a correlation between these nine IRGs and poorer OC survival.

To assess the prognostic significance of the nine IRGs in OC, Kaplan–Meier (K–M) plot analysis evaluated each gene’s impact on overall survival (OS) (Appendix A–F). Higher levels of *LRP1*, *AKT2*, *PDGFRA*, *IL27RA*, *OBP2A*, *PAEP*, *FGF7*, *FOS*, and *PI3* mRNA were linked to poor survival, with increased hazard ratios. However, a mixed scenario emerged: high levels of *AKT2*, *OBP2A*, *PAEP*, *PDGFRA*, *IL27RA*, *FOS*, and *PI3* were associated with improved survival. K–M analysis showed higher mRNA levels of all nine genes correlated with better OC prognosis, evidenced by hazard ratios for *AKT2* (HR = 1.05, *p* < 0.001), *OBP2A* (HR = 1.03, *p* = 0.05), *PAEP* (HR = 1.03, *p* = 0.005), *PDGFRA* (HR = 1.09, *p* < 0.01), *IL27RA* (HR = 1.11, *p* = 0.007), *FOS* (HR = 1.02, *p* = 0.02), and *PI3* (HR = 1.02, *p* < 0.001) (Appendix A–I).

Patients were sub-categorised into low- and high-expression groups based on median gene mRNA levels. K–M survival analysis showed that high-expression groups of *PAEP* (*p* = 0.0036) and *IL27RA* (*p* = 0.033) had better overall survival (OS) rates than low-expression counterparts. Conversely, low-expression groups of *LRP1* (*p* = 0.00031), *FGF7* (*p* = 0.016), *PDGFRA* (*p* = 0.00023), and *PI3* (*p* = 7.2 × 10^−5^) (Appendix A–F) showed improved OS compared to high-expression counterparts (Table 1). These findings underscore the complex role of IRGs in OC prognosis, highlighting the need for further research into their clinical relevance.

### 2.3. The Nine IRGs Identified as a Prognostic Marker in OC

To assess the predictive capacity of the nine IRG mRNA levels in determining OC patient survival, cases from the ICGC dataset were divided into high-risk (n = 426) and low-risk groups (n = 88) using the “maxstat” package with *p* < 0.01 (Figure 2A). OS analysis showed a significant difference, with lower risk scores correlating with better prognosis (Figure 2B, *p* < 0.01). ROC curves for 1, 3, and 5 years were plotted to assess the model’s predictive performance for OS, yielding AUC values of 0.63, 0.7, and 0.76, respectively (Figure 2C). The AUC in the OC validation set was 0.698.

We investigated whether the risk score could serve as an independent predictor for OC patients. Univariate Cox regression analysis revealed significant correlations between age (*p* < 0.002, HR = 1.021), stage (*p* = 0.038, HR = 1.367), risk score (*p* < 0.001, HR = 0.5), and OS in OC patients (Figure 2D). A nomogram was developed, incorporating risk score, pathological stage, histological grade, and age to predict 1-, 3-, or 5-year OS rates. Calibration curves for these predictions closely aligned with actual outcomes (Figure 2E–G). This investigation underscores the utility of the nine mRNA levels in the IRGs model for prognostic evaluation in OC, aiding clinical decision-making by providing insights on survival outcomes.

### 2.4. Prognostic Implications of Nine IRGs in OC

To elucidate the relationship between the nine IRGs and OC prognosis, we analysed their mRNA levels in OC and normalized them against normal tissue expression. Using the TCGA database, we compared expression levels between tumours and matched normal tissues. *LRP1*, *AKT2*, *PI3*, *IL27RA*, *OBP2A*, and *PAEP* were found to be more expressed in OC tumour samples than in normal tissues (Appendix A–F). Conversely, *FGF7*, *FOS*, and *PDGFRA* were less expressed in OC tumour samples than in normal tissues (Appendix A–I).

To predict OC outcomes using IRGs, we performed univariate Cox regression analysis on the TCGA-OC dataset, identifying 31 differential IRGs with *p* < 0.05, of which 14 had a hazardous effect on prognosis (Figure 3A). To avoid over-fitting, LASSO Cox regression analysis was then conducted (Figure 3B), resulting in the identification of nine IRGs—*LRP1*, *IL27RA*, *FGF7*, *AKT2*, *FOS*, *PAEP*, *OBP2A*, *PDGFRA*, and *PI3*—for developing an immune prognosis model (Figure 3C).

Further multivariate analysis unveiled significant insights: *LRP1*, *PAEP*, *PI3*, *OBP2A*, and *IL27RA* genes exhibited higher levels in the tumour group, whereas *FOS*, *PDGFRA*, and *FGF7* showed higher expression in normal ovarian tissue (all *p* < 0.001). Notably, *AKT2* did not display a significant difference in mRNA levels between tumour and normal groups (Figure 3D, *p* = 0.0686). Additionally, the scrutiny of protein levels of the nine model genes in normal ovarian samples and OC incidences, available from The Human Protein Atlas (https://www.proteinatlas.org/) (accessed on 15 February 2024), uncovered disparities between RNA and protein expression levels (Appendix A–I). This observation accentuates the complexity of post-transcriptional regulation and translational modifications, highlighting the need for further investigation in these areas.

### 2.5. The Nine IRGs as Indices for Diagnosing OC 

We conducted an ROC curve analysis to evaluate the diagnostic potential of nine IRGs in distinguishing between 591 OC patients and 88 normal ovarian samples. The analysis revealed significant discriminatory potential for certain IRGs, with *LRP1* showing an AUC of 0.872 and *FGF7* an AUC of 0.688, with *p*-values of 1.8 × 10^−11^ and 2.5 × 10^−2^, respectively (Figure 4A,B).

For the remaining IRGs, the AUCs varied, with *OBP2A*, *IL27RA*, *PI3*, *FOS*, *PDGFRA*, *PAEP*, and *AKT2* yielding values of 0.644, 0.621, 0.604, 0.595, 0.576, 0.545, and 0.510, respectively. However, the statistical significance varied among these IRGs, with corresponding *p*-values of 0.085, 0.099, 0.27, 0.14, 0.22, 0.34, and 0.46 (Figure 4C–I). These findings highlight the varying diagnostic utility of the individual IRGs, with *LRP1* and *FGF7* demonstrating the most promising discriminatory capabilities in distinguishing OC patients from normal ovarian samples.

### 2.6. The Nine IRGs Are Enriched to Indicate Enhanced Oncogenic Activities in OC

We analysed the nine mRNA classifier IRGs in OC through GSEA using DEGs between high- and low-risk groups. Results showed enhanced angiogenesis, indicating increased protein synthesis and active cell proliferation in the high-risk group. Moreover, activation of antifolate resistance, the cytosolic DNA-sensing pathway, and the IL17 signalling pathway, associated with OC progression, suggested a higher metastatic risk due to enhanced metastatic abilities in the high-risk group (Appendix A–D).

To comprehensively summarise the functional implications of the nine IRGs in OC progression, we utilised the CancerSEA database [33]. Functional analysis revealed significant correlations between nine IRGs and various cancer-related functional states, highlighting their complex roles in OC biology. For example, *LRP1* exhibited negative correlations with DNA repair and positive correlations with differentiation, angiogenesis, inflammation, and stemness (Appendix A). Similar correlations were observed for *AKT2*, *FGF7*, *FOS*, *IL27RA*, *OBP2A*, *PAEP*, *PDGFRA*, and *PI3*, each highlighting their involvement in diverse aspects of OC progression (Appendix A–I, Appendix A).

An AUC of ≥0.5 for all genes indicates their strong diagnostic potential for OC. This analysis highlights the importance of these nine IRGs, as their significant AUC values suggest robust diagnostic capabilities. The nine IRGs can potentially serve as valuable biomarkers for the early detection and prognosis of OC.

The nine IRGs play essential roles in regulating various cancer hallmarks such as DNA repair, cell cycle, differentiation, angiogenesis, inflammation, stemness, apoptosis, quiescence, and epithelial–mesenchymal transition (EMT) in OC. These correlations underscore their significance in OC progression, offering insights for targeted therapy and prognostic assessment.

Functional enrichment analysis investigated the roles of *LRP1*, *IL27RA*, *FGF7*, *FOS*, *AKT2*, *PAEP*, *OBP2A*, *PI3*, and *PDGFRA* in cancer based on *LRP1* mRNA levels. Results showed complex associations: high *LRP1* expression was inversely related to apoptosis and inflammation, yet positively correlated with pathways linked to coagulation, xenobiotic metabolism, and KRAS signalling (Appendix A–D). These findings suggest metabolic-related pathways influenced by high *LRP1* expression.

### 2.7. The Nine IRGs Reflect Immunosuppression in High-Risk OC Patients

We utilized the “ESTIMATE” algorithm to assess stromal and immune scores in OC samples, revealing that the high-risk group exhibited significantly higher stromal scores (526.6 vs. 337.8, *p* = 0.0139) and slightly lower immune scores (429.9 vs. 258.7, *p* = 0.0547) (Appendix A) compared to the low-risk group. This suggests a heightened stromal presence and potentially diminished immune activity in the high-risk group.

An analysis of 22 types of tumour-infiltrating immune cells showed distinct patterns between the two risk groups. The low-risk group exhibited increased infiltration of CD8+ T cells, T follicular helper cells, plasma cells, and resting mast cells, while the high-risk group showed greater infiltration of CD4+ memory resting cells, naive B cells, and M0 macrophages (Appendix A). These findings suggest a potentially immunosuppressive landscape in the high-risk group, indicating a possible inhibitory role of the nine IRGs in anti-tumour immunity within the OC context. Understanding this interplay between the tumour microenvironment (TME) and the immune system in OC is crucial for developing more effective immunotherapeutic strategies.

We next utilised the TIMER database for a comprehensive analysis to explore the intricate interactions between IRGs and immune infiltration within the OC microenvironment. We observed the following: B cells decrease when the genes *LRP1* (r = −0.315, *p* < 0.05), *FGF7* (r = −0.163, *p* < 0.001), and *AKT2* (r = −0.152, *p* < 0.05) increase. CD4+ T cells increase with higher levels of *LRP1* (r = 0.255, *p* < 0.001). CD8+ T cells increase with more *LRP1* (r = 0.631, *p* < 0.001), *FOS* (r = 0.266, *p* < 0.001), and FGF7 (r = 0.47, *p* < 0.001). Neutrophils slightly increase with more *LRP1* (r = 0.169, *p* < 0.01), PI3 (r = 0.184, *p* < 0.01), FGF7 (r = 0.184, *p* < 0.01), and *AKT2* (r = 0.137, *p* < 0.05). Macrophages significantly increase with higher levels of LRP1 (r = 0.499, *p* < 0.001), *FOS* (r = 0.209, *p* < 0.001), *FGF7* (r = 0.17, *p* < 0.01), and *PDGFRA* (r = 0.354, *p* < 0.001) (Appendix A–N). Dendritic cells decrease with more *LRP1* (r = −0.157, *p* < 0.001), *PDGFRA* (r = −0.212, *p* < 0.001), AKT2 (r = −0.144, *p =* 0.023), and *OBP2A* (r = −0.251, *p* < 0.001), but increase with more *PAEP* (r = 0.199, *p =* 0.0016) and *PI3* (r = 0.165, *p* < 0.01) (Appendix A–F). The findings suggest that the nine IRGs expressions are significantly associated with various immune infiltrating cells in OC progression.

Tumour purity is the percentage of cancer cells present in a sample of tumour tissue. Non-cancerous cells have an important role in tumour biology, and it is important to understand the roles of cancerous and non-cancerous cells in a tumour [34]. IRGs were observed to be associated with tumour purity in OC, based on an analysis with the ESTIMATE score (Table 2).

Our stromal score analysis showed a favourable link between the expressions of genes such as *LRP1*, *FGF7*, *FOS*, *PI3*, *PAEP*, and *PDGFRA* and tumour purity in our investigation of the effect of IRGs on OC’s TME. This demonstrates the intricate connection between stromal cells and gene expression (Table 3). According to our data, some IRGs have a substantial correlation with the purity of OC tumours, particularly when it comes to how they interact with stromal cells.

These results indicate that there is a favourable association between the OC and the ESTIMATE and stromal scores. The relevance of particular IRGs in relation to tumour purity and stromal cell proportions in OC samples is highlighted by both the ESTIMATE and stromal score studies, indicating the intricate interactions between gene expression and TME features.

In summary, the data highlight the importance of these IRGs in both promoting and preventing tumour progression by indicating a strong correlation between the expression level of genes such as *LRP1*, *PI3*, *PAEP*, *FOS*, *FGF7*, and *PDGFRA* and the degree of tumour immune infiltration in OC.

### 2.8. The Nine IRGs Expression Signatures Predict a Worse Response to Immunotherapy in OC

To assess the immunogenicity of OC patients, we analysed the expression of immune genes (PD1, PDL1, and CTLA4) in different groups. Figure 5A,B illustrate that the low-risk group exhibited higher expressions of CTLA4 (*p =* 0.0001), PD1 (*p* < 0.002), and PDL1 (*p =* 0.51, not significant), as illustrated in Figure 5C, indicating higher immunogenicity in the low-risk group. Additionally, we further examined the relationship between IPS scores and risk groups. Figure 5D–F demonstrate a significant increase in IPS + PD1 + CTLA4, IPS + CTLA4, IPS + PD1, and IPS alone scores in the low-risk group. Furthermore, we evaluated the correlation between mRNA characteristics and the expression levels of PD1, PDL1, and CTLA4. The results revealed a significant negative correlation between the risk score and the expression of PD1 (r = 0.2501, *p =* 0.0008), PDL1 (r = 0.2805, *p =* 0.0001), and CTLA4 (r = 0.2541, *p =* 0.0005), suggesting that OC patients in the high-risk group are less likely to benefit from immunotherapy, as demonstrated in Figure 5G–I.

ICGs are a novel target for cancer treatment development and play important roles in evading self-reactivity [35]. To further investigate the correlation between the levels of *LRP1*, *IL27RA*, *FGF7*, *FOS*, *OBP2A*, *PAEP*, *PI3*, *PDGFRA*, or *AKT2* mRNA and the extent of immune infiltration, we analysed the correlation between nine IRGs and the expression of 47 ICGs in OC progression. LRP1 exhibited a positive correlation with a broad spectrum of immune checkpoint markers, indicating its potential role in modulating immune surveillance and checkpoint pathways (Appendix A) in OC progression. In addition, *IL27RA* positively correlated with key checkpoint genes, including VEGFB and TGFB1, indicating its involvement in the regulation of immune responses (Appendix A) in OC. *FGF7* showed positive correlations with markers like ENDRB and TGFB1, suggesting its influence on immune cell infiltration and activity in the tumour milieu (Appendix A). FOS showed positive correlations with critical genes such as VEGFA and TGFB1, indicating its possible role in immune regulation and cancer progression (Appendix A). *AKT2* was positively correlated with VEGFB and CD274, among others, highlighting its potential involvement in cancer immunology and therapy resistance mechanisms (Appendix A). *PAEP* demonstrated positive correlations with various cytokines and chemokines, suggesting its role in the inflammatory response and immune cell recruitment (Appendix A) in OC. *OBP2A* positively correlated with CD274 (PDL1) and other immune markers, indicating its potential role in immune checkpoint regulation (Appendix A). *PI3* showed a positive correlation with cytokines like IL12A, pointing towards its involvement in the immune response against OC (Appendix A). *PDGFRA* is positively correlated with multiple immune checkpoints and modulation genes, suggesting its significant role in the tumour immune environment (Appendix A). These observations highlight the intricate interactions between IRGs and immune checkpoint pathways in OC progression. A deeper understanding of these relationships holds promise for advancing the development of targeted therapies aimed at modulating the immune response and potential treatment strategies.

### 2.9. The Nine IRGs Indicate a Poorer Response to Chemotherapy in OC 

To assess the effectiveness of common chemotherapeutic drugs based on the expression signature of the nine IRGs in OC progression, we performed the drug susceptibility test across the two risk groups (Table 4). The findings revealed that patients in the high-risk group demonstrated lower drug sensitivity to Cisplatin (*p =* 0.0045; Figure 6A), Etoposide (*p =* 0.0028; Figure 6B), Gemcitabine (*p =* 0.015; Figure 6C), Methotrexate (*p =* 0.0011; Figure 6D), and Lenalidomide (*p =* 0.038; Figure 6E) compared to those in the lower risk group.

Despite the resistance of the high-risk OC group to chemotherapy, as indicated by the nine IRG signatures, the drug susceptibility test still identified regimens to address this challenge. Patients in the high-risk OC group exhibited significant sensitivity to three targeted therapeutic drugs: Erlotinib (*p =* 0.011; Figure 6F), Imatinib (*p* < 0.001; Figure 6G), and Nilotinib (*p =* 0.032; Figure 6H).

### 2.10. The LRP1 mRNA Levels in Cancer Types

This study has delineated a distinct immune gene signature based on IRG for the risk stratification and targeted treatment of OC patients, offering a promising avenue for diagnostic prediction in clinical OC management. Among the nine IRGs, *LRP1* emerged as the most promising in terms of prognostic value, correlation with TMB, clinical implications for survival, independent predictive capacity, diagnostic potential, immune landscape analysis, functional implications, immunotherapy prediction, and chemotherapy response in the high-risk group of OC patients. Consequently, pan-cancer considerations have been extended to include LRP1 as a prospective theragnostic target. The comprehensive exploration of LRP1 and other IRGs may prove invaluable in clinical practice for the diagnosis and treatment of OC (Appendix A).

We examined the RNA expression levels of *LRP1* across various malignancies and their corresponding normal tissues to understand the association between *LRP1* and different cancer types. Initially, we utilised the TCGA databases to compare LRP1 mRNA levels in 26 cancer types with those in normal tissues (Figure 7A). Subsequently, leveraging an integrated database from the TCGA and GTEx datasets, we validated this analysis across 34 cancer types (Figure 7B). Relative to normal tissues, *LRP1* exhibited consistently lower expression in tumour samples of LUSC but was significantly expressed in tumour samples of KIPAN, HNSC, KIRC, OC, and PAAD.

### 2.11. High LRP1 Expression Predicts Poorer Prognosis in Rare Cancer Types

To further assess the prognostic significance of *LRP1* across different cancer types, we performed a *forest plot* analysis to examine its expression and its impact on OS across 39 cancer types using a Cox regression model. The analysis revealed that higher expression of *LRP1* predicted poorer survival in KIPAN, KICH, KIRP, LGG, BLCA, THCA, KIRC, and OC (Figure 8A). Eight of the cancer types showed a significantly increased hazard ratio (Figure 8A), while three of them significantly decreased hazard ratios with higher *LRP1* expression, and others showed no significance and adversely affected disease-free survival (DFS) in KIPAN, GBMLGG, PAAD, and LGG (Figure 8B). Furthermore, K–M survival analysis demonstrated that high mRNA levels of *LRP1* correlated with unfavourable prognosis in BLCA (HR  =  1.99, *p* = 3.0 × 10^−4^), OC (HR  =  3.39, *p*  =  2.4 × 10^−12^), LGG (HR  =  2.16, *p * =  7.8 × 10^−5^), GBM (HR  =  1.72, *p*  =  8.2 × 10^−3^), THCA (HR  =  2.56, *p*  =  0.05), KIRC (HR  =  1.72, *p* = 6.9  × 10^−4^), STAD (HR  =  1.62, *p* = 5.6  × 10^−3^), and STES (HR  =  1.40, *p*  =  0.02) (Figure 8C–J). Thus, a high expression of LRP1 predicted poor prognosis in OC, BLCA, KIRC, LGG, STAD, and STES based on both analyses.

### 2.12. The Relationship between LRP1 Expression and Immune Infiltration across Solid Tumours

Immune cells play a crucial role in regulating tumour invasion and progression in various cancers [36]. Increased tumour-infiltrating lymphocytes are valuable prognostic and predictive indices for the response to immunotherapy and the prognosis of cancer patients [37]. Therefore, we investigated the correlation between *LRP1* expression and immune infiltration in various cancers using the TIMER database [35]. The results demonstrated significant negative correlations between LRP1 levels and the infiltration levels of B cells in six cancer types, including OC (r = −0.325, *p* < 0.001), BLCA (r = −0.160, *p* < 0.05), GBM (r = −0.249, *p* < 0.001), THCA (r  = −0.173, *p*  <  0.001), KIRC (r  =  −0.188, *p*  <  0.001), and KICH (r  =  −0.188, *p*  <  0.001).

For CD4+ T cells, positive correlations were observed in five cancer types: OC (r = 0.164, *p* < 0.001), GBM (r = 0.140, *p* < 0.05), KIRP (r = 0.194, *p* < 0.05), KIRC (r = 0.287, *p* < 0.001), and KICH (r = 0.287, *p* < 0.001). For CD8+ T cells, LRP1 showed positive correlations in six cancer types: OC (r = 0.263, *p* < 0.001), BLCA (r = 0.280, *p* < 0.001), GBM (r = −0.341, *p* < 0.001), KIRP (r = 0.511, *p* < 0.001), THCA (r = 0.176, *p* < 0.001), and LGG (r = 0.111, *p* < 0.001).

Neutrophil infiltration showed positive correlation with LRP1 in eight cancer types, including OC (r = 0.338, *p* < 0.001), BLCA (r =  0.441, *p* <  0.001), GBM (r =  0.578, *p* <  0.001), KIRP (r =  0.585, *p* <  0.001), THCA (r =  0.385, *p* <  0.001), KIRC (r =  0.430, *p* <  0.001), LGG (r =  0.121, *p* <  0.05), and KICH (r =  0.218, *p* <  0.001). Macrophage infiltration also showed a positive correlation with LRP1 expression in eight cancer types, including OC (r =  0.547, *p* <  0.001), BLCA (r =  0.537, *p* <  0.001), GBM (r =  0.566, *p* <  0.001), KIRP (r =  0.675, *p* <  0.001), THCA (r =  0.461, *p* <  0.001), KIRC (r =  0.326, *p* <  0.001), LGG (r =  0.160, *p*  <  0.05), and KICH (r =  0.417, *p* <  0.001). Dendritic cell infiltration was positively correlated with LRP1 expression in seven cancer types: BLCA (r = 0.413, *p* < 0.001), GBM (r = 0.228, *p* < 0.001), KIRP (r = 0.473, *p* < 0.001), THCA (r = 0.542, *p* < 0.001), KIRC (r = 0.417, *p* < 0.001), LGG (r = 0.160, *p* < 0.05), and KICH (r = 0.374, *p* < 0.001) (Appendix A–H). These data indicate that the LRP1 expression is significantly associated with the levels of immune-infiltrating cells in OV, BLCA, GBM, KICH, and KIRP cancer.

### 2.13. Correlation between LRP1 Expression and Tumour Purity in Cancers

We investigated the relationship between LRP1 expression and tumour purity across various cancers by analysing the ESTIMATE score.

Across all cancer types, the mRNA level of LRP1 analysis showed a positive correlation with tumour purity in several cancers: KIPAN (r =   0.590, *p* <  0.001), BRCA (r =   0.570, *p* <  0.001), BLCA (r  =  0.540, *p*  <  0.001), KIRC (r  =  0.45, *p*  <  0.001), LUAD (r  =  0.450, *p* <  0.001), THCA (r  =  0.390, *p* <  0.001), KIRP (r =   0.500, *p* <  0.001), PAAD (r =  0.600, *p* <  0.001), PRAD (r =   0.370, *p* <  0.001), and OC (r  =  0.250, *p* <  0.001) (Appendix A–J).

To determine the relationship between LRP1 and stromal cells in cancer, we analysed the stromal score. The LRP1 mRNA level showed a negative association with tumour purity in LIHC (r = −0.02, *p =* 0.74), but it was positively related to tumour purity in several other cancers: BRCA (r =  0.80, *p* <  0.001), OC (r =  0.38, *p* <  0.001), PRAD (r =  0.48, *p* <  0.001), HNSC (r =  0.47, *p* <  0.001), KIRC (r =  0.51, *p* <  0.001), THCA (r =  0.52, *p* <  0.001), LUAD (r =  0.55, *p* <  0.001), BLCA (r =  0.61, *p* <  0.001), and KIPAN (r =  0.60, *p* <  0.001) (Appendix A–I).

In summary, BRCA, BLCA, OV, PRAD, KIRC, THCA, LUAD, and KIPAN showed a positive correlation with both the stromal score and the ESTIMATE score, while LIHC showed a negative correlation with these scores.

### 2.14. Functional Enrichment Analysis of LRP1 across Cancers

To better understand the potential mechanisms of LRP1 in cancer, we performed a functional enrichment analysis based on the high and low expressions of *LRP1* (Appendix A). The results showed that a high mRNA level of *LRP1* was positively associated with actin filament-based processes, amyloid beta clearance by cellular and catabolic processes, amyloid-beta clearance by transcytosis, aorta development, aorta morphogenesis, and apoptotic cell clearance, according to GO analysis. Additionally, the HALLMARK enrichment term suggested that an increased expression of LRP1 was positively associated with coagulation, complement, and epithelial–mesenchymal transition. Overall, these findings suggest that high LRP1 expression is positively involved in metabolism-related pathways.

## 3. Discussion

The integrated bioinformatics approach utilised in our study provides a comprehensive understanding of OC biology by integrating multi-omics data to unravel the intricate molecular mechanisms driving disease progression [38]. Leveraging large-scale datasets, we gained insights into the complex interactions between IRGs, the TME, and patient outcomes, laying the foundation for precision medicine approaches in OC management. OC presents a major clinical challenge due to its early-stage asymptomatic nature and high recurrence rate [39,40], with approximately 70% of patients experiencing relapse and developing resistance to chemotherapy [41]. While immunotherapy has shown promise in various cancer types, its efficacy in OC has been discouraging [42]. Recent evidence has highlighted the hyperactive TME in OC as a crucial orchestrator of tumour progression, therapy resistance, and metastasis [43]. The correlation between IRGs and the TME further complicates the understanding of OC biology [44]. Previous research identified nine IRGs that could predict the prognosis of OC, providing potential for clinical prognosis prediction [45]. Additionally, 14 key IRGs were significantly correlated with OC prognosis [46]. These findings accentuate the complex and context-dependent nature of IRG interactions with the TME and their implications for understanding and treating OC.

Our findings highlight the prognostic significance of the nine identified IRGs (*AKT2*, *FGF7*, *FOS*, *IL27RA*, *LRP1*, *OBP2A*, *PAEP*, *PDGFRA*, and *PI3*) in OC. LRP1, a large endocytosis receptor, influences multiple physiological processes and regulates cellular signalling [30]. High expression levels of both LRP1 and FGF7 were associated with poorer survival outcomes, consistent with previous reports [47,48]. In contrast, elevated mRNA levels of *AKT2*, *OBP2A*, *PAEP*, *PDGFRA*, *IL27RA*, *FOS*, and *PI3* were linked to improved survival, aligning with multiple previous research outputs [49,50,51,52,53,54]. Our study identified PI3 levels as an independent predictor for the prognosis of OC patients. However, some inconsistencies with earlier reports exist, possibly due to differences in methodology and sample size. This intricate connection between these genes and OC progression highlights their potential as valuable prognostic indicators, aiding clinicians in patient risk stratification and treatment decision-making.

LRP1 plays diverse roles in multiple biological processes, including lipoprotein metabolism, endocytosis, cell growth, cell migration, inflammation, and apoptosis [30]. It also regulates platelet-derived growth factor receptors, calcium signalling [55], and blood–brain barrier (BBB) permeability [56]. Additionally, LRP1 is implicated in conditions such as neurodegenerative diseases, atherosclerosis, and cancer [57,58,59]. Despite its varied roles, the correlation between LRP1 and the OC-TME remains largely unexplored. Our study addresses this gap, highlighting LRP1’s multifaceted role in OC. Beyond its prognostic significance, LRP1 emerges as a key regulator of the TME, influencing immune cell infiltration and immune checkpoint pathways, thereby facilitating tumour immune evasion and progression. This positions LRP1 as both a prognostic biomarker and a potential therapeutic target in OC.

Our KEGG and GSEA analyses revealed that in OC, elevated LRP1 expression is primarily associated with actin filament-based processes, amyloid beta clearance, aorta development, coagulation, and epithelial–mesenchymal transition, suggesting a pivotal role for LRP1 in tumour progression. Additionally, our study unveiled correlations between LRP1 expression and tumour purity across different cancer types, indicating its potential involvement in tumour–stromal interactions and progression. LRP1, as an active endocytosis receptor for the protease inhibitor α2MR, influences crucial cellular processes in tumour progression [60]. The combination of α2MR and LRP1 induces cell proliferation and activates the mitogen-activated protein kinase (MAPK) pathway in macrophage-like cells, a process blocked by receptor-associated protein (RAP) and MEK1/ERK1/2 pathway inhibitors [60]. Thus, LRP1 plays a critical role in the proliferation and migration of macrophage cells. The functional enrichment analysis highlighted LRP1’s involvement in various cellular processes and signalling pathways, reinforcing its significance in tumorigenesis and progression. For example, targeting the actin/tropomyosin cytoskeleton and microtubules has been shown to enhance treatment efficacy in OC and potentially overcome resistance [61]. Furthermore, the process of transcytosis, facilitating the passage of dimeric IgA antibodies through epithelial barriers, underscores LRP1’s impact on cellular dynamics [62]. Additionally, LRP1’s involvement in cytokine–cytokine receptor interactions, intricately linked with immune reactions and cancer prognosis, underscores its broader biological significance [63,64].

Using the Oncomine, GEPIA, and TIMER databases, we explored LRP1 expression across various cancer types, revealing high LRP1 expression in numerous cancers, including OC. Further investigation using the CancerSEA scRNA-seq database indicated that LRP1 has negative correlations with DNA repair processes and positive correlations with differentiation, angiogenesis, inflammation, and stemness. The distinct functional roles of LRP1 in OC may arise from inherent tumour heterogeneity [28] or the limited single-cell data available for analysis. LRP1’s implication in OC pathogenesis supports its suitability for targeted therapies to disrupt tumour-promoting pathways and bolster anti-tumour immune responses. Modulating LRP1 activity could potentially reprogram the TME, sensitising tumours to immunotherapy and improving treatment outcomes. Moreover, LRP1’s involvement in regulating chemoresistance highlights its therapeutic potential, presenting novel avenues for addressing treatment resistance in OC patients.

The growing body of evidence indicates that the diversity of immune cell composition and infiltration significantly impacts cancer prognosis [65]. Differential immune infiltration responses can lead to either tumour rejection or progression. Reduced immunosurveillance is recognised as one of the primary factors contributing to OC’s inherent chemo-resistance and poor response to immunotherapy [66,67,68,69]. For instance, previous research has demonstrated that platinum- and taxane-based chemotherapy elicits varying immune responses in patients with heterogeneous TME, potentially enhancing antitumour immunity [70]. Moreover, studies have shown that the presence of CD8+ T cells in OC is associated with prolonged survival [17,71]. Additionally, previous studies highlight the importance of considering immune cells in the diagnosis and treatment of colon cancer [72], while another study suggests that the density and distribution of immune cells can impact the prognosis of breast cancer [73]. Although the role of immune infiltration in cancer prognosis is well-documented, there is a paucity of research exploring the relationship between LRP1 and immune infiltration in OC.

Our TIMER-based investigation found a significant correlation between LRP1 levels and various immune cell types, including B cells, CD4+ and CD8+ T cells, neutrophils, macrophages, and dendritic cells across multiple cancer types. Notably, high LRP1 expression was associated with decreased B cell infiltration and increased infiltration of other immune cell types in specific cancers. These findings suggest a potential role for LRP1 in modulating the TME and tumour–immune interactions, with implications for immunotherapy response and patient outcomes. The highly heterogeneous nature of the TME in OC further underscores the potential impact of LRP1 on disease heterogeneity and patient outcomes, highlighting the need for further investigation [74,75]. The Sangerbox tool-based investigation shows the correlation between LRP1 and biomarkers such as PD1, CTLA4, CD28, VEGFA, and HAVCR2. Our analysis indicated that higher levels of LRP1 expression are positively associated with T-cell exhaustion. Since T-cell exhaustion leads to immune escape, cancer cells can evade immunosurveillance [76]. In various models and clinical trials, blocking LRP1 has been shown to enhance phagocytosis, leading to a reduction in tumour burden both in vitro and in vivo [77]. These findings suggest that cancer therapies targeting LRP1 hold promise for the treatment of OC.

### Shortcomings of the Current Study

While our findings are promising, they have not yet been validated by in vitro and in vivo models. Further studies are required to provide new insights into the mechanisms by which LRP1 influences OC progression. The association of immune-related genes (IRGs) with the prognosis of other histological types of OC has been minimally explored in current research. Our study involves the use of normal tissue samples from the GTEx project, which are not patient-matched adjacent normal from the TCGA cohort. OC includes various histological types, such as serous carcinoma (70%), endometrioid carcinoma (10%), clear cell carcinoma (5%), mucinous carcinoma (2.4%), and other tumour types, each characterised by distinct clinical and molecular features [78]. However, the data for OC is not characterised according to the histological subtypes in The Cancer Genome Atlas (TCGA) for comprehensive analysis. Additionally, current studies did not account for the impacts of several clinical factors, such as radiotherapy, chemotherapy, and targeted drug therapy [79,80,81,82,83,84]. These factors should be considered in future research to provide a more comprehensive understanding of LRP1’s role in OC and to enhance the applicability of our findings in clinical settings.

## 4. Materials and Methods

A flowchart detailing the methodology of this study is presented in Figure 9.

### 4.1. Dataset Selection 

To examine the expression of IRGs in OC, we sourced data from both the Gene Expression Omnibus (GEO) database (96 cases) and TCGA (492 cases), alongside normal (noncancerous) ovarian samples from the Genotype-Tissue Expression (GTEx) project. Specifically, we downloaded five OC expression chip datasets, GSE14407 [85], GSE14001 [86], GSE26712 [87], GSE29450 [88], and GSE66957 (https://www.ncbi.nlm.nih.gov/geo/query/acc.cgi?acc=GSE66957, accessed on 1 February 2024), from the GEO database (www.ncbi.nlm.nih.gov/geo/ accessed on 1 February 2024). After correcting for batch effects, these datasets were utilised to compare IRG expression. The mRNA-Seq transcriptome profiles of 588 OC patients were obtained from TCGA (https://tcga-data.nci.nih.gov/tcga/ accessed on 1 February 2024) and the Gene Expression Omnibus (GEO) database, and data for 167 normal ovarian tissue samples were sourced from the GTEx project (http://gtexportal.org accessed on 1 February 2024). Clinical information for these samples was obtained from the UCSC Xena platform (https://xena.ucsc.edu/ accessed on 1 February 2024 [89] Appendix A).

### 4.2. Differential Gene Analysis and Functional Enrichment of IRGs 

Differential expression genes (DEGs) were identified by comparing cancer tissue with normal (noncancerous) tissue, applying a threshold of |Log2 Fold Change (FC)| > 1 and *p*-value < 0.05, as defined by the criteria available at http://gepia2.cancer-pku.cn/#degenes accessed on 1 February 2024 [90]. To analyse the expression levels of immune genes in OC tissues, we utilised the GEO2R tool on the datasets, comparing them with levels in normal ovarian tissues. From the ImmPort database (https://www.immport.org/ accessed on 1 February 2024), we downloaded a comprehensive list of 1960 IRGs for analysis. The intersection between DEGs and IRGs involved 554 genes, implying they are differentially expressed immune-related genes (DEIRGs).

To explore the biological functions and pathways associated with the differentially expressed immune genes, we employed gene ontology (GO) [91] and Kyoto Encyclopedia of Genes and Genomes (KEGG) cluster analysis packages [91]. Cytoscape software (V3.10.2) was utilised for network visualisation and ggplot2 was used for statistical graphics to aid in the visualisation of these analyses.

The IRG expression levels were used to stratify the dataset into two groups. Subsequently, enrichment analysis was carried out to evaluate the biological importance of these IRGs using KEGG pathways and HALLMARK gene sets. Gene sets that demonstrated a substantial relationship with the biological processes and pathways important to OC were identified as significantly enriched. These gene sets included those with a Normalised Enrichment Score (NES) > 1, a nominal *p*-value < 0.05, and a False Discovery Rate (FDR) q-value< 0.25.

### 4.3. Prognostic Risk Model Construction and Validation

A calibration curve was used to evaluate the model’s accuracy, and Kaplan–Meier (K–M) and Receiver Operating Characteristic (ROC) curves were used to visualise its predictive potential. Additionally, the potential of the risk score as an independent predictor of OC survival was investigated by *univariate* and *multivariate* analyses.

### 4.4. Prognostic Potential Assessment of Key Genes in OC Patients

To assess the potential of key genes in predicting prognosis in OC patients, 4929 cases were categorised into high and low groups based on the average expression level of significant genes. The survival curve was then created using K–M survival analysis and the *R package* “survival” based on each sample’s OS status and OS time information [92]. The *log-rank test* was used to determine statistical significance [93].

### 4.5. Tumour-Infiltrating Immune Cells Fraction Calculation

CIBERSORT, a universal calculation method for quantifying cell fractions from bulk tissue gene expression profiles, was used to estimate the relative proportion of immune cells [85]. The ESTIMATE algorithm is used to infer the ratio of stromal cells and immune cells in tumour samples based on gene expression characteristics.

### 4.6. Estimation of the Immunoreactivity

Immunophenoscore (IPS) was evaluated by considering four types of molecules: MHC molecules (MHC), checkpoints or immunomodulators (CP), effector cells (EC), and suppressor cells (SC). The IPS was calculated and normalised within a range of 0 to 10 [94]. Higher scores indicated higher immunoreactivity. IPS data were downloaded from TCGA.

### 4.7. Detection of the Diagnostic Values of Risk Indicators in OC

The diagnostic values of the risk indicators for OS were assessed using ROC curve analysis based on the SPSS version 26.

### 4.8. Drug Susceptibility Testing

The Connectivity Map (CMap) database (https://portals.broadinstitute.org/cmap/) (accessed on 21 February 2024), which includes 231 chemotherapeutic drugs, was used to screen potential chemotherapy drugs. We used the *R package* “*pRRophetic*” to explore the relationship between the half-maximal inhibitory concentration (IC_50_) and the model and to evaluate drug sensitivity [32].

### 4.9. Analysis of IRG Expression and Survival Analysis in OC

Data on the expression of nine IRGs in both tumour and normal tissue were retrieved from the GTEx and TCGA datasets. Expression data were *log2* (TPM + 1) transformed for analysis. K–M analysis with the log-rank test was used to compare OS for patients stratified by median gene expression levels. A univariate Cox model was used to evaluate the link between gene expression levels and patient survival in various malignancies. Statistical significance was defined as a *p*-value < 0.05.

### 4.10. Relationship between the Expression of Nine IRGs and Immunity 

We examined the relationship between the expression of these genes and the percentage of tumour-infiltrating cells, including B cells, CD8+ T cells, CD4+ T cells, dendritic cells (DCs), macrophages, and neutrophils, using the TCGA datasets. This analysis aimed to provide a better understanding of the relationship between the nine IRGs and immunity. Additionally, the stromal score and ESTIMATE score were utilised to assess tumour purity and stroma cell proportion [95]. Furthermore, we explored the relationship between each gene expression and ICGs, tumour mutation burden (TMB), and microsatellite instability (MSI) using Sangerbox 3.

### 4.11. Statistical Analysis 

GraphPad Prism 8.0.2 was used for *student-paired t*-test and correlation analysis [95]. All graphs were generated using R 4.3, GraphPad Prism 8.0.2, and SPSS 26.0. The *two-sided student *t*-test p* < 0.05 was considered the threshold of statistical significance.

## 5. Conclusions

In this study, we have identified the gene *LRP1* as a potential prognostic marker for OCusing integrated bioinformatics approaches. Our comprehensive analysis of multiple RNA-seq databases has highlighted the significant role of LRP1 in OC models, suggesting its potential as an effective therapeutic target. Additionally, the integrated bioinformatics platform we developed can be employed to identify other potential therapeutic target genes across various cancer types.

Through our analysis, we have revealed LRP1’s significant involvement in tumour progression and various cellular processes and signalling pathways. These include actin filament-based processes, amyloid-beta clearance, aorta development, coagulation, and EMT. This multifaceted role underscores LRP1’s potential as a prognostic biomarker, therapeutic target, and key regulator of the TME. The pivotal functions of LRP1 in tumour immunity and metabolism extend its therapeutic potential beyond OC. The identification of IRGs as important biomarkers in OC represents a significant advancement towards personalised care. However, despite the promising data, the role of LRP1 in OC progression remains unexplored in both in vitro and in vivo models.

Our study highlights the necessity for further research into the molecular mechanisms underlying LRP1-mediated tumorigenesis and immune modulation. Understanding the heterogeneity of OC is crucial for developing novel diagnostic tools, prognostic markers, and targeted therapies aimed at improving outcomes for cancer patients.

## Figures and Tables

**Figure 1 ijms-25-07996-f001:**
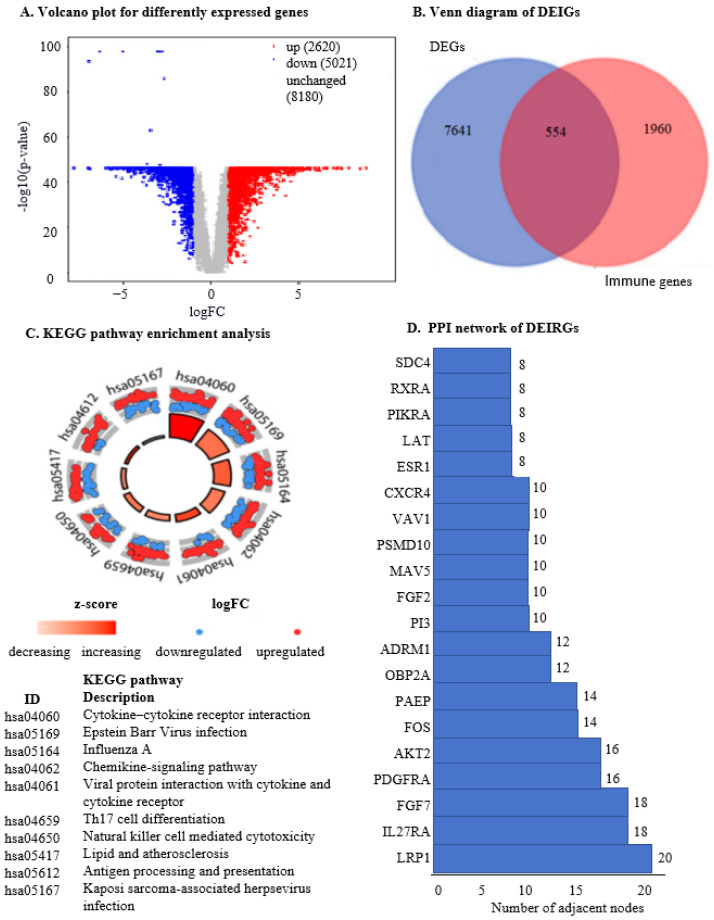
The identification and functional categorisation of differentially expressed IRGs in OC patients. (**A**). The volcano plot illustrates the differences in gene expression between tumour and normal samples. The vertical axis (y-axis) represents the mean value of −log10 (*p*-value), indicating the significance of the gene expression changes, while the horizontal axis (x-axis) represents the log fold change (*logFC*), showing the magnitude of the expression differences. Red dots denote upregulated genes, blue dots represent downregulated genes, and grey dots indicate genes with unchanged expression levels. (**B**). Venn diagram visualises the overlap between different sets of genes. Light blue indicates differentially expressed genes in patients with active OC, dark yellow represents immune genes in OC patients, and red denotes differentially expressed immune genes specific to OC patients. This helps to identify and distinguish the IRGs that are differentially expressed in the context of OC. (**C**). KEGG pathway enrichment analysis categorises the differentially expressed immune-linked genes according to their involvement in various biological pathways. The horizontal axis represents the degree value of each target, with ‘hsa’ indicating Homo sapiens (human). The red colour in the KEGG pathway indicates an increased *z-score*, suggesting pathway activation, while white indicates a decreased *z-score*. Blue dots represent downregulated genes (*logFC*), and red dots indicate upregulated genes (*logFC*), providing a comprehensive view of the pathways affected by the differentially expressed immune genes. (**D**). The protein–protein interaction (PPI) network visualises the interactions between the differentially expressed IRGs. The y-axis lists the genes, while the x-axis shows the number of adjacent nodes in increasing order, from SDC4 to LRP1. This network helps to identify key regulatory genes and their interaction partners, shedding light on the molecular mechanisms underlying immune responses in OC.

**Figure 2 ijms-25-07996-f002:**
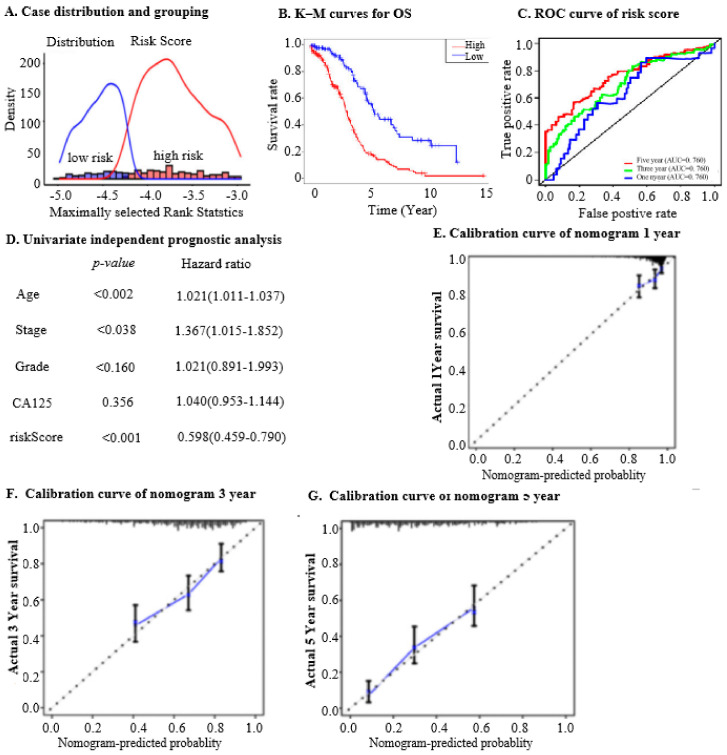
The analysis of prognostic IRGs signature and independent prognostic analysis involves evaluating the characteristics and predictive power of a set of nine prognostic genes. (**A**). Case distribution and grouping based on risk score in the test set: This chart displays the distribution of cases in the test set, categorised according to their risk scores. Patients are grouped into high-risk and low-risk categories based on their calculated risk scores, allowing for the assessment of the prognostic value of the IRGs signature. (**B**). Kaplan–Meier (K–M) curves for overall survival (OS) in high- and low-risk groups in the test set: K–M survival curves compare the OS between high-risk and low-risk groups in the test set. These curves help visualise the survival probability over time, demonstrating the impact of the risk score on patient prognosis. (**C**). Receiver operating characteristic (ROC) curve of risk score at 1, 3, and 5 years for OS in the test set: The ROC curves evaluate the predictive accuracy of the risk scores at 1, 3, and 5 years for OS. The area under the curve (AUC) indicates the effectiveness of the risk scores in predicting patient outcomes, with higher AUC values signifying better predictive performance. (**D**). Univariate independent prognostic analysis: This analysis assesses the prognostic significance of individual factors, including the nine prognostic IRGs. By evaluating each factor independently, this analysis identifies which genes or clinical variables are significantly associated with OS. (**E**). Calibration curve of nomogram at 1 year: The calibration curve at 1 year compares the predicted probabilities of OS with the actual observed outcomes. This visualisation assesses the accuracy of the nomogram in predicting 1-year survival rates, indicating how well the model’s predictions align with real-world data. (**F**). Calibration curve of nomogram at 3 years: Similar to the 1-year calibration curve, this graph compares the predicted and observed probabilities of OS at 3 years. It helps evaluate the long-term predictive accuracy of the nomogram for medium-term prognosis. (**G**). Calibration curve of nomogram at 5 years: This calibration curve extends the assessment to 5 years, comparing predicted and observed survival probabilities to evaluate the model’s accuracy for long-term prognosis. It provides insight into the reliability of the nomogram for extended survival predictions.

**Figure 3 ijms-25-07996-f003:**
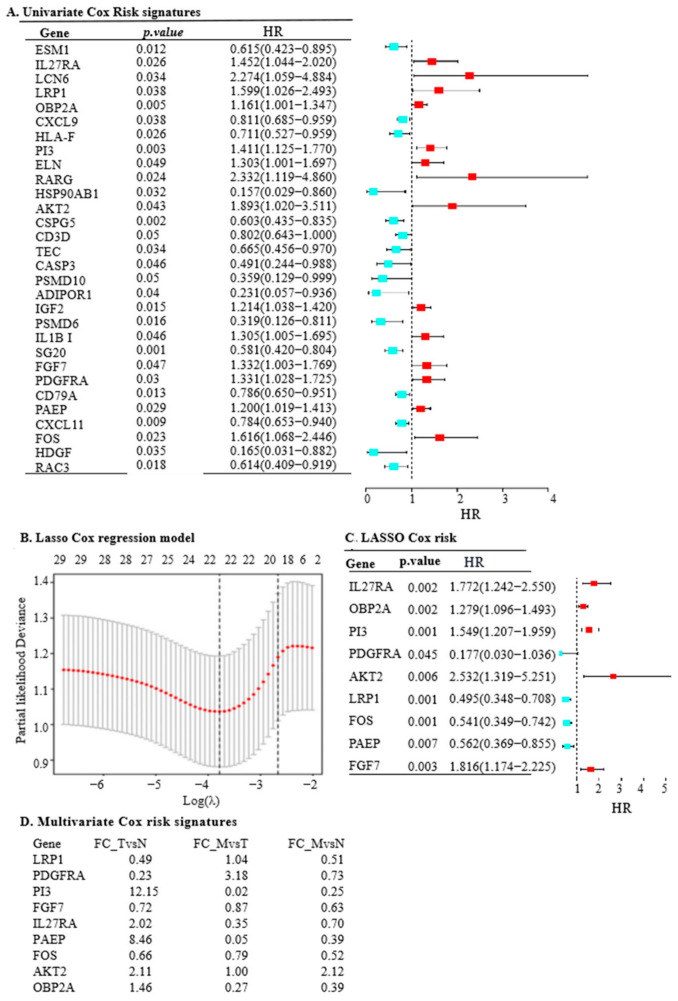
The construction of OS risk prognostic models in OC involves a series of analytical steps to evaluate and validate the prognostic significance of specific gene signatures. (**A**). Univariate Cox regression analysis of the risk signatures for OS: This analysis examines the relationship between each risk signature and OS in OC patients. By evaluating each gene or risk factor separately, the analysis identifies which signatures are significantly associated with patient survival outcomes. (**B**). LASSO Cox regression model for OS: The LASSO Cox regression model is used to refine the selection of prognostic genes. In this model, the x-axis represents the logarithm of the penalty parameter (*log-lambda*), while the y-axis denotes the partial likelihood of deviance. This model helps to minimise overfitting by applying a penalty to the number of variables, ensuring that only the most significant genes are included in the final prognostic model. (**C**). LASSO Cox regression model analysis of nine prognostic IRGs: This step involves applying the LASSO Cox regression method specifically to the nine identified prognostic genes. The model helps determine the optimal set of genes that contribute most significantly to OS, providing a robust predictive signature for OC prognosis. (**D**). Multivariate Cox risk signatures for expression of nine IRGs in normal, tumour, and metastatic groups: This analysis evaluates the expression levels of the nine model IRGs across different sample groups—normal, tumour, and metastatic tissues. The fold changes are calculated for the following: FC_TvsN, fold change between tumour and normal tissues; FC_MvsT, fold change between metastatic and tumour tissues; and FC_MvsN, fold change between metastatic and normal tissues. The analysis also incorporates the Least Absolute Shrinkage and Selection Operator (LASSO) method to further refine the gene selection.

**Figure 4 ijms-25-07996-f004:**
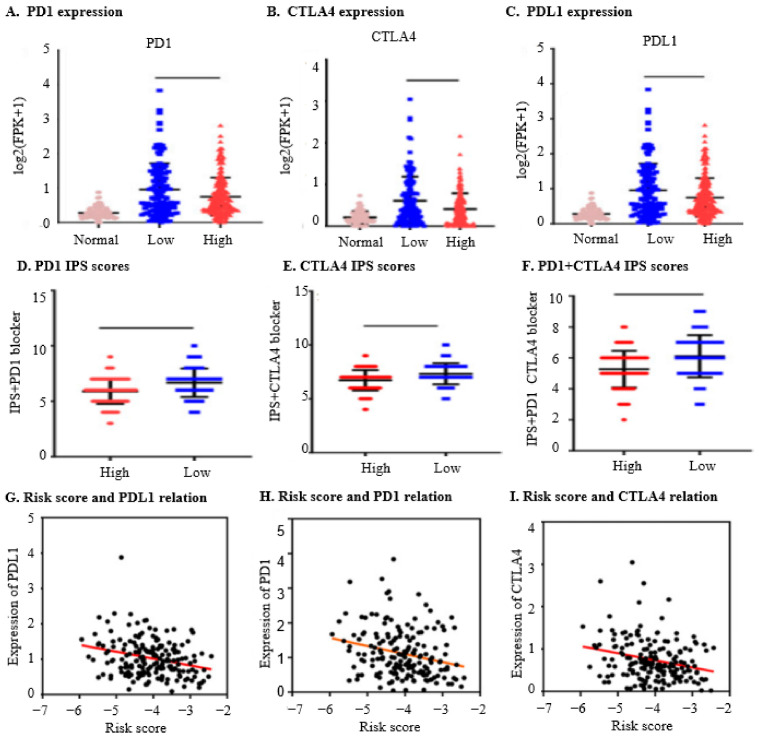
Estimation of the Immunoreactivity. (**A**). Expression of PD1 in high- and low-risk groups: The expression levels of PD1 (Programmed cell death protein 1) are measured in log2(FPKM + 1). The x-axis categorises the expression levels into three groups: normal (crimson), low (blue), and high (red) compared to OC-free groups. (**B**). Expression of CTLA4 in high- and low-risk groups: The CTLA4 (Cytotoxic T-lymphocyte-associated protein 4) expression levels are also measured in log2(FPKM + 1). The x-axis categorises the expression levels into normal (crimson), low (blue), and high (red) compared to OC-free groups. (**C**). Expression of PDL1 in high- and low-risk groups: PDL1 (Programmed death-ligand 1) expression levels, shown in log2(FPKM + 1), are categorised similarly on the x-axis into normal (crimson), low (blue), and high (red) compared to OC-free groups. (**D**). Scores of IPS and IPS + PD1 in high- and low-risk groups: The y-axis indicates the Immunophenoscore (IPS) scores, with categories on the x-axis for high-risk (red) and low-risk (blue) groups. This helps assess the immune response potential in both risk groups when considering PD1 expression. (**E**). Scores of IPS and IPS + CTLA4 in high- and low-risk groups: The IPS scores are again shown on the y-axis, with the x-axis differentiating high-risk (red) and low-risk (blue) groups, now considering CTLA4 expression. (**F**). Scores of IPS, IPS + PD1 + CTLA4 in high- and low-risk groups: The y-axis shows the IPS scores, while the x-axis indicates high-risk (red) and low-risk (blue) groups considering the combined expression of PD1 and CTLA4. (**G**). Relation of risk score and expression of PDL1 in the high-risk group: This panel depicts the correlation between risk scores (ranging from −7 to −2) and PDL1 expression levels in the high-risk group. (**H**). Relation of risk score and expression of PD1 in the high-risk group: This panel shows the correlation between risk scores (ranging from −7 to −2) and PD1 expression levels in the high-risk group. (**I**). Relation of risk score and expression of CTLA4 in the high-risk group: This panel illustrates the correlation between risk scores (ranging from −7 to −2) and CTLA4 expression levels in the high-risk group. These analyses comprehensively examine the expression of immune checkpoint genes (PD1, CTLA4, and PDL1) across different risk groups and their correlation with risk scores.

**Figure 5 ijms-25-07996-f005:**
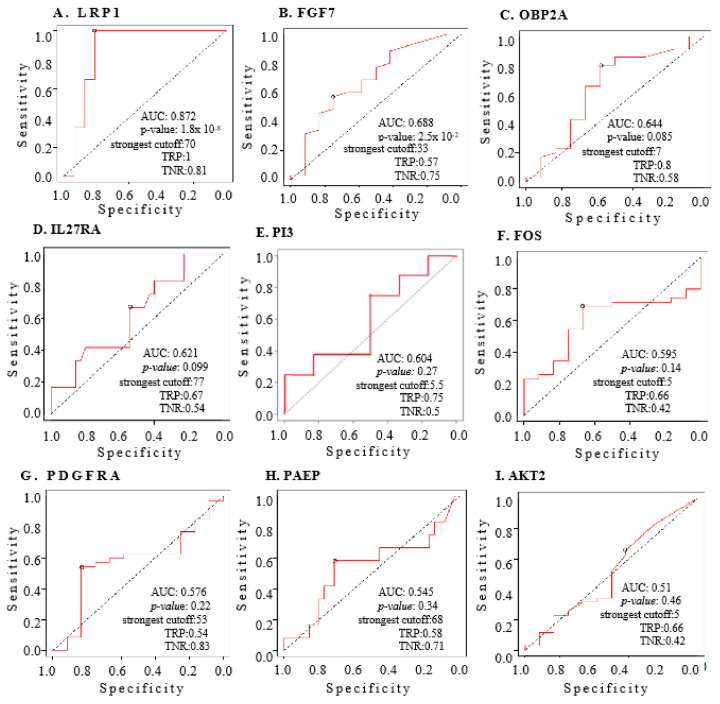
The predictive power of nine IRGs in OC was assessed using receiver operating characteristic (ROC) curve analysis, with areas under the curve (AUC) and *p*-values indicated. The ROC curve evaluates the diagnostic value of each gene by measuring its sensitivity and specificity, thereby determining its effectiveness in distinguishing between different states of the disease. (**A**). AUC of risk signature with diagnostic value of LRP1: The AUC for LRP1 (Low-density lipoprotein receptor-related protein 1) demonstrates its diagnostic value in OC. A higher AUC value indicates a stronger predictive ability for patient outcomes based on LRP1 expression levels. (**B**). AUC of risk signature with diagnostic value of FGF7: The diagnostic value of FGF7 (Fibroblast Growth Factor 7) is evaluated through its AUC, reflecting its accuracy in predicting OC presence and progression. (**C**). AUC of risk signature with diagnostic value of OBP2A: The AUC for OBP2A (Odorant Binding Protein 2A) indicates how well this gene can discriminate between OC states, contributing to its potential use as a diagnostic marker. (**D**). AUC of risk signature with diagnostic value of IL27RA: IL27RA (Interleukin 27 Receptor Subunit Alpha) is assessed for its diagnostic value, with the AUC showing its capability to predict OC outcomes effectively. (**E**). AUC of risk signature with diagnostic value of PI3: The AUC for PI3 (Peptidase Inhibitor 3) measures its diagnostic power in OC, indicating its potential role as a predictive biomarker. (**F**). AUC of risk signature with diagnostic value of FOS: The diagnostic value of FOS (Fos Proto-Oncogene) is evaluated through its AUC, highlighting its effectiveness in predicting disease states in OC. (**G**). AUC of risk signature with diagnostic value of PDGFRA: PDGFRA (Platelet-Derived Growth Factor Receptor Alpha) is assessed for its diagnostic potential, with the AUC demonstrating its predictive accuracy for OC. (**H**). AUC of risk signature with diagnostic value of PAEP: The AUC for PAEP (Progestagen Associated Endometrial Protein) indicates its diagnostic value, reflecting its sensitivity and specificity in OC prediction. (**I**). AUC of risk signature with diagnostic value of AKT2: The diagnostic value of AKT2 (AKT Serine/Threonine Kinase 2) is evaluated, showing significant sensitivity and specificity across two databases.

**Figure 6 ijms-25-07996-f006:**
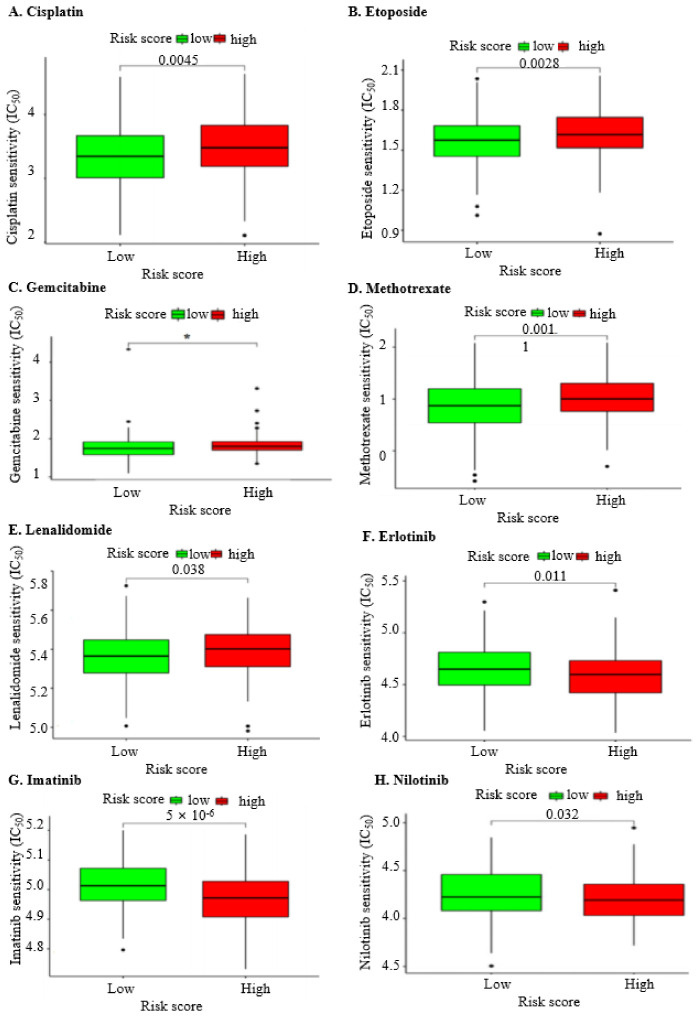
Drug susceptibility testing results. (**A**). Lower sensitivity to Cisplatin: In high-risk groups, chemotherapeutics with lower sensitivity were screened by estimating the IC_50_ values for Cisplatin. The results indicated a higher IC_50_, signifying reduced susceptibility to this drug. (**B**). Lower sensitivity to Etoposide: Similarly, lower sensitivity to Etoposide was observed in high-risk groups, as evidenced by higher IC_50_ values, suggesting decreased effectiveness of this chemotherapeutic agent. (**C**). Lower sensitivity to Methotrexate: Methotrexate also demonstrated lower sensitivity in high-risk groups, with higher IC_50_ values indicating diminished drug efficacy. (**D**). Lower sensitivity to MK-2206: The IC_50_ estimation for MK-2206 revealed lower sensitivity in high-risk groups, marking it as less effective for these patients. (**E**). Lower sensitivity to Lenalidomide: Lenalidomide showed higher IC_50_ values in high-risk groups, denoting lower sensitivity and reduced therapeutic potential. (**F**). Higher sensitivity to Erlotinib: Conversely, Erlotinib was found to be more effective in high-risk groups, with lower IC_50_ values indicating higher sensitivity to this chemotherapeutic. (**G**). Higher sensitivity to Imatinib: Imatinib also exhibited higher sensitivity in high-risk groups, as reflected by its lower IC_50_ values, suggesting better efficacy. (**H**). Higher sensitivity to Nilotinib: Lastly, Nilotinib showed greater effectiveness in high-risk groups, with lower IC_50_ values indicating increased sensitivity to this drug. These drug susceptibility testing results provide critical insights into the varying effectiveness of different chemotherapeutics in high-risk groups. Statistical significance: * indicates *p*-value < 0.05.

**Figure 7 ijms-25-07996-f007:**
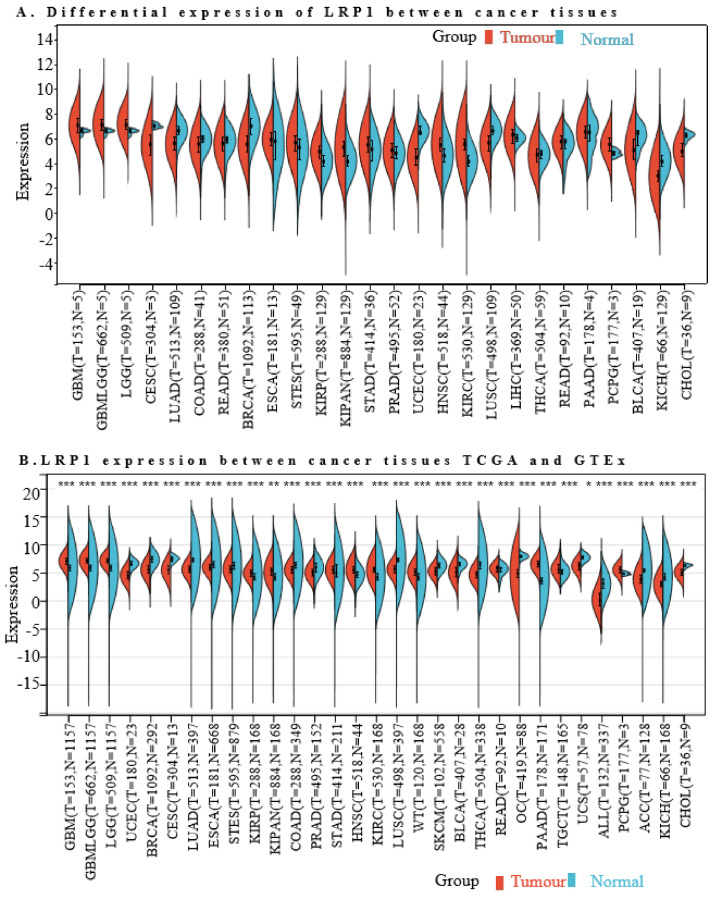
The expression of LRP1 across various cancers. (**A**). Differential expression analysis using TCGA database: The expression levels of LRP1 were compared between cancerous and normal tissues across twenty different types of cancers using data from The Cancer Genome Atlas (TCGA). This analysis provided insights into the differential expression patterns of *LRP1*, revealing significant variations in its expression between tumours and normal tissues. (**B**). Integrated analysis using TCGA and GTEx datasets: A comprehensive analysis incorporating data from both the TCGA and the Genotype-Tissue Expression (GTEx) projects was conducted for thirty-four cancer types. The results indicated that LRP1 was highly expressed in tumour samples of KIPAN (kidney cancers), HNSC (head and neck squamous cell carcinoma), KIRC (kidney renal clear cell carcinoma), OC (ovarian cancer), and PAAD (pancreatic adenocarcinoma). In contrast, *LRP1* expression was lower in lung squamous cell carcinoma (LUSC) tumour samples compared to normal tissues in both datasets. In the visual representation, red denotes LRP1 expression in tumour groups and blue represents its expression in normal groups. Statistical significance: * indicates *p*-value < 0.05; ** indicates *p*-value < 0.01; and *** indicates *p*-value < 0.001.

**Figure 8 ijms-25-07996-f008:**
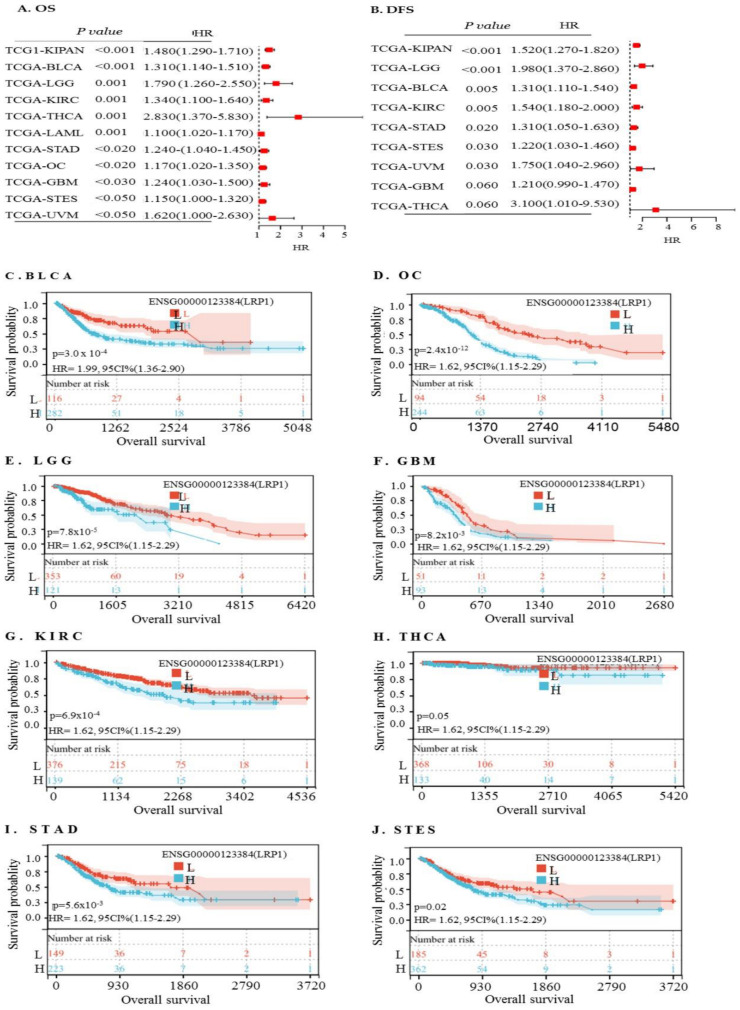
Association of high LRP1 expression with poor prognosis in cancers. (**A**). Analysis of OS using the Cox regression model: A forest plot was generated to analyse the impact of LRP1 expression on OS across 39 different cancer types. This statistical analysis provided insights into the association between high *LRP1* expression levels and worse OS outcomes in various cancers. (**B**). Analysis of disease-free survival (DFS) using the Cox regression model: Similarly, a forest plot table was constructed to examine the effect of *LRP1* expression on DFS across the same 39 cancer types. This analysis allowed for the assessment of the relationship between elevated *LRP1* expression and reduced DFS rates in different cancer cohorts. (**C**–**J**). Kaplan–Meier curve analysis of high *LRP1* expression: K–M curve analysis was performed to further elucidate the prognostic significance of high *LRP1* expression. The analysis revealed that elevated *LRP1* expression predicted poor prognosis in several cancer types, including bladder urothelial carcinoma (BLCA), ovarian cancer (OC), lower-grade glioma (LGG), glioblastoma multiforme (GBM), kidney renal clear cell carcinoma (KIRC), thyroid carcinoma (THCA), stomach adenocarcinoma (STAD), and stomach and oesophageal carcinoma (STES).

**Figure 9 ijms-25-07996-f009:**
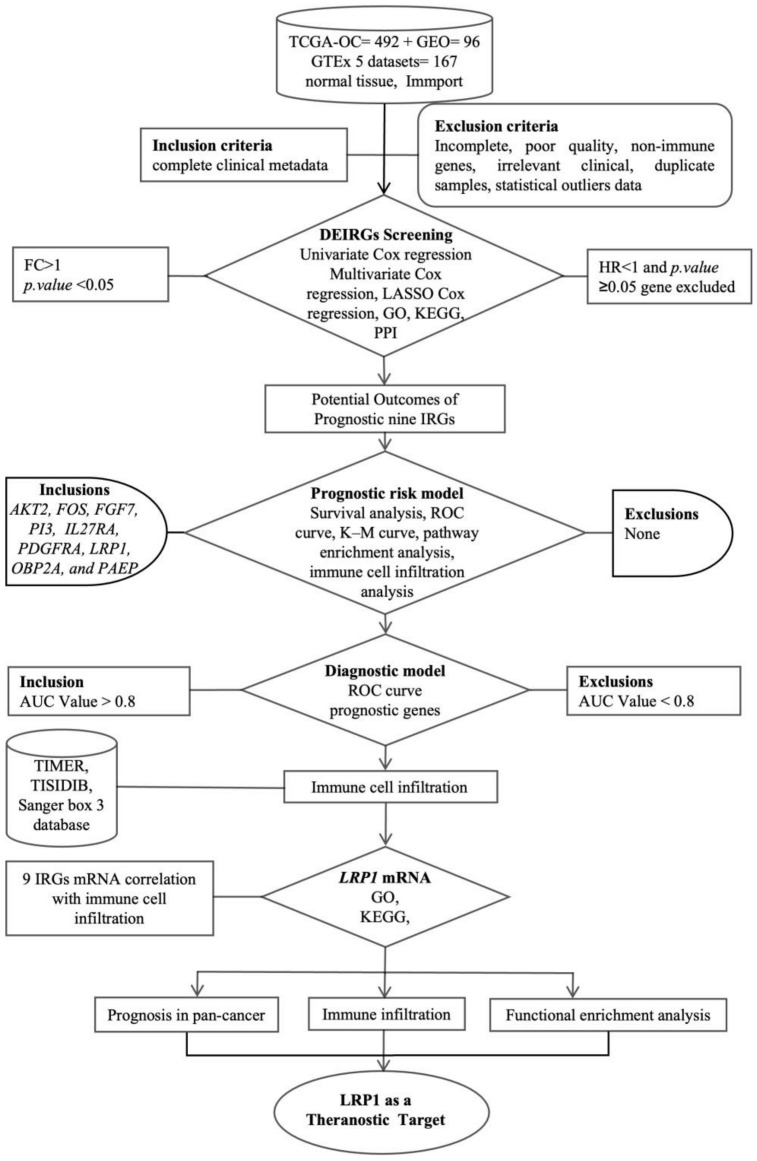
The flowchart provides a schematic overview of the study design, illustrating the impact of immune-associated gene expression on the prognostic, therapeutic, and diagnostic identification, and validation of these genes in ovarian cancer (OC). The study integrates several analytical approaches and databases to achieve a comprehensive understanding of the immune landscape in OC. Gene Ontology (GO) and Kyoto Encyclopedia of Genes and Genomes (KEGG) analyses are utilised to categorise and map the functional roles and pathways of the differentially expressed immune-related genes (IRGs). Tumour mutation burden (TMB) and microsatellite instability (MSI) metrics are assessed to evaluate the genetic alterations and their implications on immune responses. The TIMER (Tumor Immune Estimation Resource) and TISIDIB (tumour–immune system interaction) databases are utilised to estimate the extent of immune cell infiltration within the tumour microenvironment. The Sanger Box 3 databases are referenced for further insights into immune cell infiltration patterns. This integrative approach facilitates a robust evaluation of the prognostic potential of IRGs, enhances diagnostic accuracy, and identifies potential therapeutic targets in OC.

**Table 1 ijms-25-07996-t001:** Impact of gene expression levels on OS rates in patients.

Gene	*p*-Value	HR (95% CI)	Figure
*PAEP*	0.0036	0.80 (0.69–0.93)	Appendix A
*IL27RA*	0.033	0.86 (0.76–0.99)	Appendix A
*LRP1*	0.00031	1.28 (1.12–1.47)	Appendix A
*FGF7*	0.016	1.18 (1.03–1.36)	Appendix A
*PDGFRA*	0.00023	1.32 (1.14–1.53)	Appendix A
*PI3*	7.2 × 10^−5^	1.34 (1.16–1.55)	Appendix A

Footnote: This table illustrates the impact of different gene expression levels on overall survival (OS) rates in patients. The hazard ratios (HR) with 95% confidence intervals (CI) indicate the risk of death associated with each gene’s expression level, where values above one suggest increased risk and values below one suggest decreased risk. The *p*-values demonstrate the statistical significance of these findings. Referenced figures provide visual representations of these associations.

**Table 2 ijms-25-07996-t002:** The ESTIMATE score correlation between IRGs and tumour purity, with their corresponding correlation coefficients, and *p*-values.

Gene	Correlation Coefficient	Correlation Coefficient (r)	*p*-Value	Figure
*LRP1*	Positive	0.34	8.9 × 10^−13^	Appendix A
*PI3*	Positive	0.19	8.7 × 10^−5^	Appendix A
*PAEP*	Positive	0.2	4.0 × 10^−5^	Appendix A
*FOS*	Positive	0.21	1.1 × 10^−5^	Appendix A
*FGF7*	Positive	0.66	7.4 × 10^−54^	Appendix A
*PDGFRA*	Positive	0.46	9.9 × 10^−24^	Appendix A

Footnote: This table highlights the positive correlations between IRGs and tumour purity, as indicated by the ESTIMATE scores. The correlation coefficients (r) and *p*-values highlight the strength and significance of these relationships. The figures referenced provide visual confirmation of these statistical associations.

**Table 3 ijms-25-07996-t003:** The correlation between IRGs and tumour purity, with their corresponding correlation coefficients, *p*-values, and figures for reference.

Gene	Correlation Coefficient (r)	*p*-Value	Figure
*LRP1*	0.49	1.2 × 10^−26^	Appendix A
*FGF7*	0.81	7.4 × 10^−97^	Appendix A
*FOS*	0.28	6.5 × 10^−9^	Appendix A
*PI3*	0.12	0.02	Appendix A
*PAEP*	0.19	9.9 × 10^−5^	Appendix A
*PDGFRA*	0.64	6.1 × 10^−49^	Appendix A

Footnote: The *p*-values and correlation coefficients (r) in this table illustrate the strength and significance of the relationship between IRGs and tumour purity. The figures referenced provide visual representations of these correlations, reinforcing the statistical findings.

**Table 4 ijms-25-07996-t004:** Drug susceptibility test results among low and high-risk groups.

Drug	Risk Group		*p*-Value
	Low	High	
Cisplatin	3.3	3.52	0.0045
Etoposide	1.56	1.65	0.0028
Gemcitabine	1.78	1.83	0.0011
Methotrexate	0.9	1.04	0.0011
Lenalidomide	5.37	5.41	0.038
Erlotinib	4.6	4.56	0.011
Imatinib	5.04	4.97	5 × 10^−6^
Nilotinib	4.2	4.12	0.032

Footnote: The *p*-values presented in this table were calculated using a *two-sided Student’s t*-test to determine the statistical significance of the differences in drug susceptibility between low and high-risk groups. A lower *p*-value indicates a higher statistical significance, suggesting that the observed differences in drug response are less likely to be due to chance.

## Data Availability

The datasets generated, coded, and analysed during the current study are available on request.

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
