# Peer review of "The Integrated Bioinformatic Approach Reveals the Prognostic Significance of LRP1 Expression in Ovarian Cancer"

_ijms, 2024, doi:10.3390/ijms25147996_

Round 1

Reviewer 1 Report

Comments and Suggestions for Authors

The study aimed to implement bioinformatic approach for understanding prognostic significance of immune genes specially LRP1 in ovarian cancer. The authors have performed very comprhensive anlayses using variety of cancer related data. The introduction is well-written, providing sufficient literature to support the study's hypothesis. The statistical methods used in the analysis are well-justified. The presentation of results could be improved, though the discussion is thorough. Here are my comments:

Line 775: Are the 167 samples adjacent normal or blood normal? Please specify.

Line 798-799: 554 IRGs were downloaded from ImmPort, which includes only a few cancer-related studies. Why? There are several other immune studies performed at the pan-cancer level (example - https://gdc.cancer.gov/about-data/publications/panimmune), which provide a more comprehensive list of immune-related genes.

Line 795: What is the p-value threshold for identifying differentially expressed genes?

Paragraph "Tumour-Infiltrating Immune Cells Fraction Calculation..": The text of this paragraph has discontinuity. Please check.

Figure 2a: Line 151, -log10 (adjusted p-value), In the figure, it is p-value. Be consistent.

Figure 2b: The Venn diagram shows total immune genes as 554+1960; however, you mentioned that only 554 IRGs were downloaded from the database.

Line 174: What is SI 2F? Is it a supplementary image?

Line 180: SI 4A-I, not all the figures are in the supplementary file. Some figures are not as mentioned in the text. Please check these inconsistencies.

Line 205-207: The text doesn't match the figure. Where are these two groups shown in the figure?

Table 2: Can you provide the actual p-values?

Line 425: Have you analyzed nine or three immune genes for understanding the response to immunotherapy? Line 428 mentions only three genes, while the figure mentions all nine.

Author Response

Dear Reviewer,

Reviewer 2 Report

Comments and Suggestions for Authors

In this paper, the authors carried out an exhaustive examination of OC data to investigate the function and heterogeneity of Immune-related genes (IRGs) via data mining, where nine IRGs were identified, with potential prognostic and diagnostic values examined. I mainly have one major concern below:

Given the mixed and vague signals for the nine IRGs and the fact that LRP1 has been identified in previous literature, I suggested the authors compare using LRP1 alone to using all nine IRGs when examining their prognostic and diagnostic values, such as the analysis in Section 2.3.   

Author Response

Dear Reviewer, 

Round 2

Reviewer 1 Report

Comments and Suggestions for Authors

Comment#1 I see that TCGA contains only 492 cases (transcriptome) of TCGA-OV (https://portal.gdc.cancer.gov/projects/TCGA-OV), but you mentioned 588. Did you get data from other resources? Please calrify. Second, the 167 normal samples are not adjancent normals of these TCGA's patients. These normal samples were taken from GTEX. In that case, how will you justifty the individual's specific heterogenity. The confounding factors such as age or gender were also not considered. The cancer sample and adjacent normal from the same patient is recommended to analyse and that provide more reliable results.  

Comments on the Quality of English Language

Most of my comments were addressed except one which I have mentioned to authors. 

Author Response

Dear Reviewer, 

We thank you for highlighting errors and constructive suggestions for the manuscript.

Please find the attachment of the response to your comment. 

Reviewer 2 Report

Comments and Suggestions for Authors

The authors have addressed my concerns.

Author Response

Dear Reviewer, 

 We thank you for your time. 

Round 3

Reviewer 1 Report

Comments and Suggestions for Authors

All comments were addressed